# Investigating the barriers and enablers to outbreak reporting in the Asia-Pacific region: A mixed-methods study protocol

Amish Talwar[1]*, Rebecca Katz[2], Martyn D. Kirk[1], Tambri Housen[1,3]

**1** National Centre for Epidemiology and Population Health, Australian National University, Canberra, Australian Capital Territory, Australia, **2** Center for Global Health Science and Security, Georgetown University, Washington, DC, United States of America, **3** School of Medicine and Public Health, University of Newcastle, Newcastle, New South Wales, Australia

* amish.talwar@anu.edu.au

**Data Availability Statement:** No datasets were generated or analysed for the creation of the protocol. Once the study proposed in this protocol paper is completed, all relevant data generated

## Abstract

### Introduction

The COVID-19 pandemic has raised concerns about the global capacity for timely outbreak reporting. However, gaps remain in our understanding of barriers and enablers to outbreak reporting, particularly at the local level. Field epidemiology training program (FETP) fellows often participate in the outbreak reporting process as part of both their training and the public health roles they assume after graduating; they therefore represent a potentially valuable source of information for better understanding these barriers and enablers. This study will investigate the barriers and enablers to outbreak reporting through a mixed methods approach that will encompass a review of the existing literature as well as surveying and interviewing FETP trainees and graduates from the Asia-Pacific region.

### Methods

This study will begin with a scoping review of the literature to identify existing evidence of barriers and enablers to outbreak reporting. Based on our findings from the scoping review, we will administer a survey to FETP trainees and graduates from the World Health Organization Western Pacific and South-East Asian Regions and conduct interviews with a subset of survey respondents to investigate the survey findings in more detail. We will summarise and compare the survey results according to various country-level economic and political indicators, and we will employ thematic analysis to evaluate the interview responses. Based on the findings from the scoping review, survey, and interviews, we will construct a model to comprehensively describe the various barriers and enablers to outbreak reporting.

### Conclusion

This study will contribute to our understanding of the determinants of outbreak reporting across several geographic, political, and economic contexts by eliciting the viewpoints and experiences of persons involved with outbreak reporting, particularly at the local level. This

through that study will be made available upon study completion through the Australian Data Archive.

**Funding:** The author(s) received no specific funding for this work.

**Competing interests:** The authors have declared that no competing interests exist.

information will help improve the outbreak reporting process, allowing for more timely reporting and helping prevent future outbreaks from becoming pandemics.

## Introduction

The global reaction to the COVID-19 pandemic has raised concerns about the world's collective ability to detect, report, and respond to infectious disease outbreaks. Outbreak reporting is the process by which infectious disease outbreaks, once detected, are reported to public health officials; initial reporting to local public health officials is typically followed by escalation of the report to more senior public health authorities and, where necessary, to the international community to formulate an appropriate outbreak response. Timely outbreak reporting in turn can make the difference between containing an outbreak at its source and an outbreak growing into an epidemic or pandemic. The timeliness of outbreak reporting is contingent on a state's capacity to detect and report an outbreak up to the national level and its willingness to report it to the international community once aware of the outbreak within its borders. Thus, under the 2005 International Health Regulations (IHR), all IHR signatories have an obligation to develop and maintain the capacity to detect infectious disease outbreaks that can become international threats and to report these outbreaks to the international community through the World Health Organization (WHO) [1]. Under the IHR, most countries have agreed to report potential health events of international significance within 24 hours of determining the existence of such an event [1].

However, delays in outbreak reporting at various levels of the reporting chain have persisted even after the revised IHR came into effect. This was the case for the Ebola outbreak in Western Africa and most recently during COVID-19, where local delays in reporting the existence and extent of the COVID-19 outbreak at its initial stages delayed the global response to the outbreak by several weeks, allowing it to grow into a global pandemic [2, 3]. To prevent future pandemics, it is crucial to understand why nearly 20 years following the adoption of the most recent version of the IHR countries still experience failures reporting outbreaks in line with their international obligations.

Although there is a wide breadth of research on the various barriers and enablers across the outbreak reporting chain, comparatively little research has focused on the barriers and enablers at the local level, where outbreak reports originate. This is particularly true in the Asia-Pacific region, from where several pandemics have emerged in recent decades. Field epidemiology training programs (FETPs) produce skilled epidemiologists who learn to perform public health functions through experiential learning in field settings at the local and subnational level [4]. As part of their training, FETP trainees typically participate in surveillance and outbreak reporting activities [5]. Thus, FETP trainees and graduates have unique insights into the various determinants of outbreak reporting, particularly at the local and subnational levels, including the routine operations of reporting systems. Therefore, this study seeks to leverage the knowledge of FETP-trained officials in the Asia-Pacific region to fill this knowledge gap by examining their observations and experiences regarding the factors that inhibit or enable outbreak reporting through a survey and key informant interviews.

### Evidence gap

Previous studies that have examined the determinants of outbreak reporting have largely focused on countries' ability to detect and report outbreaks. While outbreak preparedness

indices such as the Global Health Security Index (GHSI) mainly report on nation-level capacity for outbreak detection and reporting, other studies have focused on determinants commonly found across different countries. In one such study, officials from Brazil, Ethiopia, Liberia, Nigeria, and Uganda identified a series of bottlenecks and enablers to outbreak detection and notification at all levels of their national public health systems [6, 7]. These included the availability of core surveillance infrastructure (including laboratory infrastructure), physical infrastructure (including access to appropriate communications and transportation to facilitate outbreak detection and reporting), personnel resources and knowledge, community knowledge, and inter-agency cooperation and coordination [7]. Having adequate and appropriately trained personnel to report remains a lynchpin for outbreak reporting. Therefore, additional research has examined the role of health care providers at the local level in reporting, who often are the first to detect potential outbreak cases and have the responsibility to report these cases to public health officials. These studies have identified various barriers to reporting among health care providers, including lack of reporting knowledge as well as lack of time and motivation to report [8–12].

Although capacity to report is a prerequisite for timely outbreak reporting, an environment conducive to reporting is also important for effecting an appropriate outbreak response, especially for outbreaks that threaten to grow rapidly and cross borders. A handful of studies have examined this additional condition for reporting among national governments. Two such studies involved surveys and interviews with officials affiliated with National Focal Points (NFPs), which are the country persons or offices responsible for collecting outbreak information and relaying information on potentially significant outbreaks to WHO for further dissemination to the international community to ensure an appropriate global response [13]. While the respondents agreed that inadequate surveillance and reporting infrastructure was a major reporting barrier at all levels, they also indicated that fear of damage to tourism and trade as well as "political challenges" were important barriers to outbreak reporting at the national level [13, 14]. These findings provided qualitative evidence supporting the results of a previous study, which found that countries that are particularly vulnerable to trade or travel barriers after reporting an outbreak or in which higher domestic political opposition is present were more likely to demonstrate less timely reporting internationally, even after controlling for capacity to detect an outbreak [15].

Although these studies have helped define the determinants of outbreak reporting at various levels among a handful of countries, few studies have investigated the experiences of local and subnational public health officials within the Asia-Pacific region. This represents a significant gap given that this region contains over half the world's population, and four of the seven respiratory disease pandemics of the 20th and 21st centuries emerged from this region alone [3, 16–20]. The handful of studies from the region that have investigated reporting barriers and enablers at the local and subnational levels have largely corroborated the above findings; however, they have focused on specific countries or disease contexts [21–23]. Furthermore, none of these studies have examined in-depth whether reporting officials, including both public health officials and health care providers in public and private settings, have experienced pressures to not report for either political, security, or economic reasons, despite such pressures likely having played a role during previous outbreaks, including SARS and COVID-19 [24, 25]. Lastly, these studies have failed to assess the relative importance of the various putative barriers and enablers to outbreak reporting. To address these gaps, this study will survey and interview FETP trainees and graduates across the Asia-Pacific region on their experiences and observations regarding putative barriers and enablers to outbreak reporting. As the first known region-wide mixed methods study on outbreak reporting, it will examine these barriers and enablers across the varying political, economic, and developmental contexts that define

the Asia-Pacific region. In turn, it will inform regionally relevant approaches to improve outbreak reporting and collaborative response.

## Study question and objective

Our research question is the following:

• What are the barriers and enablers to outbreak reporting in the Asia-Pacific region at the local and subnational levels?

The specific objectives of this study will be to elicit the beliefs of FETP trainees and graduates from the Asia-Pacific region on the relative importance of putative barriers and enablers to outbreak reporting in a survey and to elaborate on these findings based on the personal knowledge, beliefs, and experiences of a select group of survey respondents through in-depth interviews. This information will help identify potential interventions that can improve the outbreak reporting process and help prevent future outbreaks from becoming pandemics.

## Methods and analysis

### Study design

This study will employ a cross-sectional mixed methods approach to assess these barriers and enablers by using both quantitative and qualitative methods for collecting data, which will then be triangulated to seek corroboration or divergence between the quantitative and qualitative findings and to derive findings and insights not possible with either approach alone [26]. Specifically, this study will use an explanatory sequential design (Fig 1) in which a survey will first be administered to FETP trainees and graduates to explore their views on the relative importance of putative barriers and enablers to outbreak reporting, followed by in-depth interviews with selected respondents to provide further insights into the data generated by the survey [26, 27]. The survey questions will be informed by a scoping review of the literature and consultations with subject matter experts affiliated with FETPs in the Asia-Pacific region and will be piloted with persons experienced with FETPs and outbreak reporting. Following survey administration, we will conduct semi-structured interviews with a subset of survey respondents to investigate the survey findings in more detail and to evaluate how they differ among varying geographic, political, and socioeconomic contexts [26, 28]. Thus, the interviews will focus on both explaining and contextualizing the survey findings based on the unique perspectives and circumstances of these participants. Using data from the scoping review, survey, and interviews, we will construct a candidate conceptual model to describe the various barriers and enablers to outbreak reporting from the local level to the national level and internationally. Participant recruitment began on 16 November 2023 and remains ongoing; we intend to end recruitment on 31 May 2024.

### Study population

Our study population for the survey will be persons with outbreak reporting responsibilities in the WHO Western Pacific Region (WPRO) and South-East Asian Region (SEARO) [29]. The

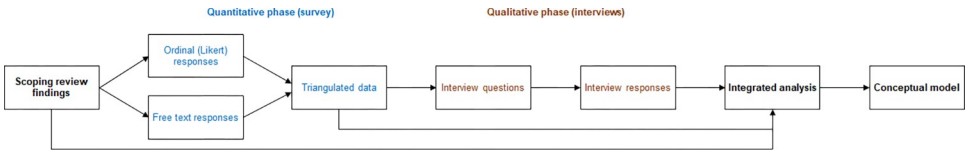

**Fig 1. Explanatory sequential study design.**

nature and size of the population of persons with outbreak reporting responsibilities at the local and subnational levels in the WHO WPRO and SEARO regions are unclear, as is the most appropriate means to recruit among this population. Therefore, we will employ criterion sampling, in which we will specifically recruit FETP trainees and graduates, a readily identifiable subset of this population [28]. Specifically, we will approach all FETPs from the two regions to administer the survey among current trainees and graduates to capture a wide range of regional backgrounds and responses. In addition, we will disseminate the survey through the Training Programs in Epidemiology and Public Health Interventions Network (TEPHINET) global alumni listserv. TEPHINET helps develop and connect the various FETPs globally and can reach FETP alumni registered with TEPHINET [30].

For the interview phase, we will approach a subset of survey respondents who indicate willingness to be interviewed. We will employ maximum variation sampling to select a diverse range of interviewees to reflect viewpoints and experiences across a variety of reporting settings among the regions surveyed [28]. We will preferentially target survey respondents who provide detailed free text responses.

## Sampling

**Survey.** As of 2023, there are 3,623 graduates from the 21 FETPs in the WPRO and SEARO regions listed with TEPHINET combined [31]. It is unclear how many FETP participants are currently in training; based on our experience with FETPs in these regions, we estimate that each of the 21 WPRO and SEARO FETPs listed with TEPHINET typically has two to five trainees at any one time, for a total of approximately 42–105 trainees from both regions [31]. For this study, we will use the mean of this range (74 trainees) as our estimate for the total number of trainees. As part of field epidemiology training, all trainees are expected to be involved in surveillance and outbreak reporting. Although it is unclear how many FETP graduates from these regions continue to perform outbreak- and surveillance-related activities after graduating, a previous study found that 65.7% of graduates from the Eastern Mediterranean region investigate outbreaks, and 69.9%–71.7% work in public health surveillance [5]. Assuming that approximately 70% of graduates in the WPRO and SEARO regions also conduct surveillance and outbreak reporting activities, we anticipate a sample frame for this study of 2,610 persons. Based on a Cochran sample size calculation, we estimate our target sample size to be 335 persons at $\alpha = 0.05$ and $p = 0.5$ [32]. Previous studies have found the response rate for online surveys to be between 34% and 48% [33]. Because of the added challenges of responding to an online survey in a resource-limited setting and potential language barriers in completing a survey in the English language, we will assume a more conservative response rate of less than half this range, meaning that we assume a minimum 17% response rate; for our target sample size, this would require reaching out to 1,971 persons. Thus, to meet our sample size goal, we will reach out to all 21 listed FETPs in the WPRO and SEARO regions to obtain permission to circulate the survey to trainees and graduates. We will also solicit participation from FETPs not listed with TEPHINET to maximize our sample. To determine a more exact target sample size for this population, we will ask each training program to provide information on the total number of trainees and graduates within their programs.

**Semi-structured interviews.** We will select respondents using a maximum variation sampling approach to reflect the geographical and economic diversity among the WPRO and SEARO countries and to identify shared patterns of experiences that cut across different national settings [28]. Interviews will continue until saturation has been reached (i.e., new data repeats what was expressed in previous data, and elicited themes begin to stabilise) or the list of candidate interviewees has been exhausted [34]. We will aim to interview at least 20 persons,

with no more than three persons representing any individual country to ensure representation of varying reporting contexts [35].

## Data collection

**Scoping review.** We will scope both the peer-reviewed and grey literature according to the PRISMA Extension for Scoping Review guidelines (PRISMA-ScR) to develop a more comprehensive understanding of the existing evidence of barriers and enablers to outbreak reporting among both primary data (e.g., data collected through interviews, focus groups, and surveys) and secondary data analysis (e.g., analysis of country-level data) [36]. We will identify relevant peer-reviewed literature (both quantitative and qualitative studies, including systematic and scoping reviews) and grey literature (non-peer reviewed reports and government documents) published from 15 January 2008 (six months after the 2005 IHR went into effect) to 31 December 2023 among three online databases (PubMed, Scopus, and Web of Science) and Google Scholar. We will use search terms designed to elicit sources that address 1) outbreak reports and 2) barriers and enablers to outbreak reporting. In addition, we will search the reference lists of sources chosen for data abstraction for additional literature to review, and we will review country outbreak After Action Review reports provided by WHO [37].

**Survey.** The survey will collect demographic information as well as experiences with FETPs and outbreak reporting. Demographic information will include age, gender, level of education, country of work, public health role, type of FETP training, and country through which the FETP training was completed. The questions investigating knowledge and experiences with putative barriers and enablers to reporting will be generated based on previous research, including the scoping review. These questions will focus on barriers and enablers related to capacity, coordination, and communication; training and socialization around reporting; motivation and incentives to report; and authority to report. In addition, the survey will investigate whether reporting officials have felt pressure to not report or feared economic consequences from reporting to evaluate the impact of organizational, political, and economic factors on reporting. The relative importance of these barriers and enablers to outbreak reporting will be assessed using a Likert scale of 1 to 3, where 1 equals "No impact", 2 equals "Some impact", and 3 equals "High impact". Free text prompts will allow respondents to further contextualize their responses. We will refine the survey based on input provided by subject matter experts affiliated with FETPs in the Asia-Pacific region. We will then pilot the survey in Australia with four subject matter experts experienced with FETPs and outbreak reporting in the Asia-Pacific region, including FETP trainers and graduates, to assess content validity of the survey questions. The survey will be administered through Qualtrics [38].

We will analyse the survey findings in the context of various empirical economic and political indicators by which the respondents' country of work can be classified. Although a previous study did not find a correlation between regime type and outbreak reporting time, a later study did find that domestic commitment to rule of law was significantly associated with outbreak reporting time [15, 39]. This latter study also found that development level (i.e., gross domestic product [GDP] per capita), exports and imports as a share of GDP, and contribution of travel and tourism to GDP were significantly associated with outbreak reporting time [15]. Another study found that increased internet usage and freedom of the press were associated with timelier outbreak reporting, which might reflect the availability of unofficial channels for outbreak reporting, including rumours and media reports [40]. To account for the impacts of these factors on outbreak reporting, we will stratify the respondents' country of work according to the following covariates and compare the survey results across these strata: income level, trade exposure according to GDP, contribution of travel and tourism to GDP, rule of law

based on the Worldwide Governance Indicators, number of internet users, and freedom of the press [41–45].

**Semi-structured interviews.**   Interviews will be either held in-person or online on Zoom between the first author and each participant individually. The interviews are anticipated to last for 45 to 60 minutes. The results of the scoping review and survey will inform the creation of semi-structured interview questions for the interviews. These questions will elicit further contextual information and perspectives regarding commonly identified barriers and enablers as well as details about unique barriers and enablers that are outliers with respect to these findings. We will pilot the interview in Australia with at least one subject matter expert experienced with FETPs and outbreak reporting in the Asia-Pacific region, and we will transcribe the interviews verbatim to facilitate data analysis.

**Reflexivity.**   The primary researcher's background as a physician and public health specialist with field epidemiology training and outbreak response experience informed his interest in the research question based on his experience that timely outbreak detection and reporting are crucial to effective outbreak containment. This background also helped the primary researcher identify FETP trainees and graduates as a likely valuable source of information on the outbreak reporting process. However, the primary researcher's training and experience can also affect the interview questions asked, interpretations made, and the researcher-interviewee dynamic itself by providing a pre-defined understanding of outbreak reporting for the researcher and how it can be ideally realised, which might differ from the experiences and understandings of the interviewees. To facilitate reflexivity, the primary researcher will keep a research journal to record post-interview reflections to engage their evolving perception of the data with respect to their own experiences, shape understanding of the researcher-interviewee dynamic, inform future interviews, and refine candidate interview themes [46].

## Data analysis

**Scoping review.**   Data extraction and analysis will proceed per established guidelines for scoping reviews [47]. We will extract the following data from the selected literature into a Microsoft Excel spreadsheet: study author, title, journal name, year of publication, and study type; study purpose, methodology, target population, location, and start/end date; and evidence for barriers and enablers to outbreak reporting. Based on the extracted data, we will summarize the reviewed literature based on numbers of sources per country, target population, study type, and methodology; we will present these data in a table along with a descriptive summary. We will also identify common findings across the various literature reviewed and will synthesize these thematically into a coherent narrative description, which will be summarized in a table or flowchart. The results of this review will inform the construction of survey questions about respondents' experiences with barriers and enablers to outbreak reporting.

**Survey.**   We will report counts and percentages for all variables. For Likert variables, we will report median and interquartile range. We will analyse differences in weighted median ordinal responses and GHSI scores for detecting public health events according to the covariates described above (rule of law, trade exposure, contribution of travel and tourism to GDP, country income level, number of internet users, and freedom of press), where weighted medians will be calculated for each stratum [48]. We will compare weighted median ordinal responses and GHSI scores between different income groups using the Kruskal-Wallis test, and we will employ the pair-wise Wilcoxon rank sum test for post-hoc analysis to determine which groups significantly differ ($\alpha = 0.05$). For rule of law, trade exposure, travel and tourism, internet use, and freedom of press indicators, we will assess for association with weighted

median ordinal responses and GHSI scores using Spearman's rank correlation ($\alpha = 0.05$). We will perform all quantitative survey analyses in R [49].

We will thematically summarize the findings from the free text box responses using content analysis, where we will inductively code the text into thematic categories and triangulate these findings with the scaled responses [27, 50]. We will then use the triangulated findings to inform the interview questions, thereby integrating the study designs [50, 51]. Where example quotes are provided verbatim, the quotes will not be attributable by country to protect respondents from unintentional identification.

**Semi-structured interviews.** We will use thematic analysis incorporating the Framework Method to analyse the interview data [52, 53]. First, we will deductively code themes elicited from the findings from the scoping review and survey and group the codes within a working analytical framework [53]. Using these codes, two researchers will then conduct a thematic analysis employing a "coding reliability" approach where they will label themes identified in the interviews with these predetermined codes; while one of the coders will be an existing researcher on our team, the other coder will be an outside researcher unaffiliated with the study [52, 53]. To begin, the coders will review the interview transcripts and corresponding audio to familiarise themselves with the data. Next, the two coders will code the interview data into a framework matrix [53]. After coding at least the first three transcripts, the two researchers will meet to compare their coded transcripts and to adjust the analytical framework where necessary given the transcript data, including adding new codes or modifying/deleting existing codes [53]. The coders will continue to adjust the analytic framework as needed until the final transcript has been coded [53]. On completion of coding, the coders will interpret the matrix data to identify overarching themes, including themes consistent with the previous scoping review and interview data as well as novel ones [53]. In addition, we will calculate the level of agreement between the two coders using Cohen's kappa and recode as necessary if significant differences emerge [52]. Where necessary to discuss participant work location, we will report WHO region and country income level stratum instead of country name to protect interviewee privacy given that both interview findings and country name data might together facilitate interviewee identification. We will code the interview data using NVivo 12 and generate the analytical framework within a Microsoft Excel spreadsheet [54].

**Data interpretation.** We will depict the integrated findings from the scoping review, survey responses, and interviews using joint displays to visually draw out unique insights that are only accessible through joint interpretation of the quantitative and qualitative data [51, 55]. Based on these findings, we will develop a conceptual model incorporating all the known factors that impact outbreak reporting at the local, subnational, and national levels and their interactions across all levels. We will share preliminary interview findings with interviewees to obtain feedback on our results and interpretation.

**Patient and public involvement.** As stated above, we will consult with subject matter experts to develop the survey instrument and will reach out to TEPHINET and FETPs to disseminate the survey among FETP trainees and graduates. We will directly recruit interviewees among survey respondents who volunteer to be interviewed.

## Ethics and dissemination

### Ethics

The proposed research activities have been approved by the Human Ethics Office at the Australian National University (protocol number 2023–196). We will apply for any protocol amendments with this office.

### Consent

Before beginning the survey or interview, participants must read a "Participant Information Sheet" that outlines the study; they must then click "Yes" to the survey question asking for consent to participate in the survey or sign and submit by email or post a written consent form to be interviewed. Participation in the survey and interviews will be voluntary, and participants are free to withdraw from participation at any point while taking the survey or being interviewed.

### Confidentiality

We will keep participant identities confidential as far as allowed by law. For the survey we will not require name or contact information unless the respondent would like to be contacted to participate in an interview. We will use this information to create a candidate interviewee list; after abstracting this contact information into this list, we will delete this information from the survey platform. The data interviewees provide will be de-identified, including name, and will be redacted of any incidentally identifying information provided during the interview, such as information on specific locations or events that could be linked back to the interviewees or their countries of work. Identifying details (i.e., name and email address) will be stored separately from the rest of the research data in the candidate interviewee list, which will be linked back to each interview within this document. Access to the data will be restricted to the research team. Published results will only be reported in aggregate (except for de-identified quotations where appropriate), and participants will not be identifiable within published outputs. Furthermore, as interview participants could potentially be identified from a publication based on being associated with their country affiliation, these country affiliations will not be referred to in any publication.

### Expected output

We will use previous research and results from this study to develop a conceptual model incorporating all the known factors that impact outbreak reporting at the local, subnational, and national levels. This will allow for the coordinated development of key interventions appropriate for each reporting level to improve the outbreak reporting process. More generally, this study will highlight the need to account for the many different determinants of outbreak reporting, including capacity to detect and report outbreaks and political or economic barriers to reporting.

### Dissemination

We will share the results of this study at academic conferences and through peer-reviewed reports published in relevant research journals. All survey and interview participants will be provided a URL link to review study outputs. All study data will be retained and securely stored for at least five years following publications arising from this research. After the storage period, de-identified study data will be archived at the Australian Data Archive for use in later research, including potentially by other researchers [56].

## Discussion

In the wake of the COVID-19 pandemic, the world has begun the process of reforming the body of global health law that informs pandemic preparedness and response, including amending the 2005 IHR and adopting a new pandemic treaty [57, 58]. The proposed reforms mainly target country-level barriers to outbreak reporting, particularly the capacity to detect

and report outbreaks. However, all outbreak reports start at the local level before moving up through various layers of a country's public health system. As such, it will be crucial to evaluate and address the barriers at all levels of the outbreak reporting system to effect successful reforms to prepare for future outbreaks.

With outbreak reporting responsibilities at various levels of the public health system, FETP-trained officials represent a potentially invaluable source of information on the various barriers and enablers to outbreak reporting at all levels of a public health system. By evaluating the experiences of FETP trainees and graduates in the Asia-Pacific region, this study will build on previous studies that have evaluated outbreak reporting within specific country settings or among specific reporting groups to develop a more comprehensive overview of the various outbreak reporting barriers and enablers and to inform relevant approaches to improve reporting and collaborative response.

## Limitations

This study is subject to several limitations. Selection bias might affect the survey results, where persons with reliable internet access and fluency in the survey language will be more likely to complete the survey, potentially skewing results towards countries or areas with better internet infrastructure and personnel with English fluency. In addition, this study will elicit the views and experiences of a select population of public health officials, which might constrain the applicability of the study findings beyond settings encountered by FETP-trained officials. This study mitigates these concerns by distributing the survey in a platform that is readily accessible among various mobile phone and computer devices, including in resource-limited environments, and by recruiting persons with FETP experiences at different levels of government across various countries. Furthermore, potential respondents might choose not to participate in the survey or interviews for fear of being identified and associated with their responses, leading to harmful professional, economic, or legal impacts. To mitigate privacy concerns, the survey and interviews will be fully anonymous, unless a participant asks to be identified.

This study is also limited by the study investigators' particular knowledge and experience. As such, the questions to be asked in the survey and interviews might not adequately account for the variety of challenges faced by FETP officials from a variety of different settings. To help address this bias, we will review and pilot this survey with subject matter experts who have experience with outbreak reporting in various countries in the Asia-Pacific region. In addition, the interview research journal will help to illuminate any biases with respect to the interviewer's questioning.

Despite these limitations, this study will fill a major gap in our understanding of the determinants of outbreak reporting across several geographic, political, and economic contexts by eliciting the viewpoints and experiences of persons with exposure to outbreak reporting across various settings. This information will help improve outbreak reporting processes by informing appropriate interventions, such as targeted reporting capacity improvements, regulatory and governance changes to streamline reporting, and education among reporting and other key government officials about the importance of outbreak reporting. Such interventions will allow for more timely reporting, helping prevent outbreaks from growing into devastating epidemics or pandemics.

## Acknowledgments

We thank Amy E. Parry for her invaluable recommendations while developing this project.

## Author Contributions

**Conceptualization:** Amish Talwar.

**Methodology:** Amish Talwar, Rebecca Katz, Martyn D. Kirk, Tambri Housen.

**Supervision:** Rebecca Katz, Martyn D. Kirk, Tambri Housen.

**Writing – original draft:** Amish Talwar.

**Writing – review & editing:** Rebecca Katz, Martyn D. Kirk, Tambri Housen.

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
