## [Decision Letter · Decision Letter 0]

26 Jun 2024

PONE-D-24-06613Investigating the barriers and enablers to outbreak reporting in the Asia-Pacific region: a mixed-methods study protocolPLOS ONE

Dear Dr. Talwar,

Thank you for submitting your manuscript to PLOS ONE. After careful consideration, we feel that it has merit but does not fully meet PLOS ONE’s publication criteria as it currently stands. Therefore, we invite you to submit a revised version of the manuscript that addresses the points raised during the review process.

We look forward to receiving your revised manuscript.

Kind regards,

Olushayo Oluseun Olu

Academic Editor

PLOS ONE

Reviewers' comments:

Reviewer's Responses to Questions

**Comments to the Author**

1. Does the manuscript provide a valid rationale for the proposed study, with clearly identified and justified research questions?

Reviewer #1: Yes

Reviewer #2: Yes

Reviewer #3: Yes

2. Is the protocol technically sound and planned in a manner that will lead to a meaningful outcome and allow testing the stated hypotheses?

Reviewer #1: Yes

Reviewer #2: Yes

Reviewer #3: Yes

3. Is the methodology feasible and described in sufficient detail to allow the work to be replicable?

Reviewer #1: Yes

Reviewer #2: Yes

Reviewer #3: Yes

4. Have the authors described where all data underlying the findings will be made available when the study is complete?

Reviewer #1: Yes

Reviewer #2: No

Reviewer #3: Yes

5. Is the manuscript presented in an intelligible fashion and written in standard English?

Reviewer #1: Yes

Reviewer #2: Yes

Reviewer #3: Yes

6. Review Comments to the Author

You may also provide optional suggestions and comments to authors that they might find helpful in planning their study.

Reviewer #1: While publishing study protocols is part of the commitment to improving research standards by promoting transparency, reducing publication bias, and enhancing the reproducibility of study design and analysis. Study protocols for proposed or ongoing prospective clinical research should be those that provide a detailed account of the hypothesis, rationale and methodology of the study, and the associated ethical requirements.

The proposed study and the methodology proposed is not knew and while this study my present good findings for the Asia-Pacific region, it has been done elsewhere as cited by the authors in Bochner AF, Makumbi I, Aderinola O, Abayneh A, Jetoh R, Yemanaberhan RL, et al.467 Implementation of the 7-1-7 target for detection, notification, and response to public health threats in five countries: a retrospective, observational study. Lancet Glob Health.

Importantly this is a short duration study, and the authors are encouraged to quickly conduct the study and submit the results as a research article.

Finally, I suggest that this article is changed into a short viewpoint with the plans for answering key questions in the Asia Pacific Region

Reviewer #2: Dear Authors,

Thank you for submitting your manuscript, "Investigating the barriers and enablers to outbreak reporting in the Asia-Pacific region: a mixed-methods study protocol," to PLOS ONE. The topic is timely, and the methodology is well-structured, addressing a critical gap in our understanding of outbreak reporting dynamics in a key geographical area. Below are some comments aimed at further strengthening the manuscript before publication:

1. Clarification of Mixed Methods Design: The manuscript effectively outlines the use of mixed methods but could benefit from more detailed explanations on how integrating these methods will enhance the understanding of the study topic beyond what could be achieved by quantitative or qualitative methods alone.

2. Detailed Justification for Geographical Focus: While the Asia-Pacific region is a critical area for study, a more detailed justification for selecting this region, especially considering its diverse political, economic, and health systems, would be beneficial. This could help clarify the expected generalizability of the study findings.

3. Expansion on FETP's Role: The role of Field Epidemiology Training Program (FETP) trainees and graduates is highlighted; however, expanding on how their unique perspectives specifically contribute to understanding barriers and enablers could further enrich the narrative.

4. Methodological Rigor: More information on the validity and reliability of the survey instruments and interview protocols would be useful. Additionally, discussing any pilot testing of the instruments might help reassure readers of the methodological rigor.

5. Ethical Considerations: The manuscript mentions ethical approvals and consent processes but could further discuss how potential ethical issues in dealing with sensitive outbreak data will be managed, especially in interviews.

6. Limitations: I appreciate the discussion of limitations; however, it might be beneficial to elaborate on how these limitations could affect the study's outcomes and the steps taken to mitigate them.

7. Implications for Policy and Practice: While the manuscript discusses the study's potential impacts, specific recommendations for policy-makers and public health officials based on anticipated findings could make the conclusions more actionable.

Reviewer #3: Title – Appropriate

Abstract – Well presented. However, replace Discussion with Conclusion

Introduction – Statement of problem, magnitude of problem, rationale for the study and study objectives well presented.

Line 128 - Does this mean health care providers in public and private health facilities?

Methods and analysis – Clearly presented. Specific study design not mentioned - Is it a descriptive cross-sectional study?

Study setting, target population, sampling techniques and data management are well described

Line 155 - It is desirable to state where the piloting will be done.

Line 250 - Please indicate the number of respondents targeted for the pilot survey. Also indicate geographical locations of the pilot survey.

Line 323 - It is not clear if this will be Key Informant Interview or In-Depth Interview. It needs to be clearly stated.

Discussion – Inappropriate for proposal as a sub-section. Most of the content is Limitation of the study. This can be made the sub-section title.

References - Adequate

7. PLOS authors have the option to publish the peer review history of their article (what does this mean?). If published, this will include your full peer review and any attached files.

Reviewer #1: No

Reviewer #2: **Yes: **Sylvester Maleghemi

Reviewer #3: **Yes: **Prof. Tanimola Makanjuola Akande

---

## [Author Response · Author response to Decision Letter 0]

29 Jul 2024

Dear Editor:

We thank you and the reviewers for your thorough review of our manuscript. Regarding your concerns, we edited the document to conform with PLOS ONE editorial standards. Furthermore, we would like to clarify that as this is a protocol paper, there will be no data generated in association with it. All data will be generated through the study proposed in the protocol paper, which has not been completed yet. We clarify this point in the “Questionnaire” section as per the below:

“No datasets were generated or analysed for the creation of the protocol. Once the study proposed in this protocol paper is completed, all relevant data generated through that study will be made available upon study completion through the Australian Data Archive.”

Please find below our responses to the reviewer feedback. Note that page numbers refer to the corrections made in the manuscript with track changes.

Reviewer #1:

"While publishing study protocols is part of the commitment to improving research standards by promoting transparency, reducing publication bias, and enhancing the reproducibility of study design and analysis. Study protocols for proposed or ongoing prospective clinical research should be those that provide a detailed account of the hypothesis, rationale and methodology of the study, and the associated ethical requirements.

The proposed study and the methodology proposed is not knew and while this study my present good findings for the Asia-Pacific region, it has been done elsewhere as cited by the authors in Bochner AF, Makumbi I, Aderinola O, Abayneh A, Jetoh R, Yemanaberhan RL, et al.467 Implementation of the 7-1-7 target for detection, notification, and response to public health threats in five countries: a retrospective, observational study. Lancet Glob Health."

We respectfully disagree that this is not a novel study or methodology. The cited study by Bochner, et al. (2023) entailed a synthesis of outbreak response performance metrics (i.e., time to detection, reporting, and response) for specific outbreak events and official insights into the bottlenecks and enablers for achieving those metrics through desk reviews and workshops. Our study entails a survey of putative outbreak reporters, particularly trained field epidemiologists, to delineate the relative importance of event-agnostic barriers and enablers for outbreak reporting, followed by interviews with a subset of those respondents to obtain detailed qualitative data to further elaborate on those findings. Furthermore, our study seeks to engage a more comprehensive body of reporting barriers and enablers, including infrastructural, bureaucratic, political, economic, and personal/behavioural barriers and enablers, which is particularly needed given the paucity of literature with respect to non-technical barriers and enablers. Therefore, this represents a completely different methodology with a more focused topic of interest (outbreak reporting) while exploring a broader array of putative barriers and enablers. We clarify our study’s unique insights in lines 141-143.

"Importantly this is a short duration study, and the authors are encouraged to quickly conduct the study and submit the results as a research article.

Finally, I suggest that this article is changed into a short viewpoint with the plans for answering key questions in the Asia Pacific Region"

We also respectfully disagree that our study findings can be communicated in a short viewpoint, which is more appropriate for articulating brief opinions on current global public health issues. This is particularly true for mixed methods studies, which provide extensive quantitative and qualitative information subject to significant explication and interpretation that can only be achieved in a full scientific manuscript.

Reviewer #2:

"Dear Authors,

Thank you for submitting your manuscript, "Investigating the barriers and enablers to outbreak reporting in the Asia-Pacific region: a mixed-methods study protocol," to PLOS ONE. The topic is timely, and the methodology is well-structured, addressing a critical gap in our understanding of outbreak reporting dynamics in a key geographical area. Below are some comments aimed at further strengthening the manuscript before publication:

1. Clarification of Mixed Methods Design: The manuscript effectively outlines the use of mixed methods but could benefit from more detailed explanations on how integrating these methods will enhance the understanding of the study topic beyond what could be achieved by quantitative or qualitative methods alone."

We further elaborate on how integrating these methods will enhance understanding of the study topic in lines 161-162 and lines 165-167. Further information is communicated in the Study design section.

"2. Detailed Justification for Geographical Focus: While the Asia-Pacific region is a critical area for study, a more detailed justification for selecting this region, especially considering its diverse political, economic, and health systems, would be beneficial. This could help clarify the expected generalizability of the study findings."

We further elaborate on the justification for evaluating the Asia-Pacific region in lines 141-143, in addition to the information previously provided in the Evidence gap section.

"3. Expansion on FETP's Role: The role of Field Epidemiology Training Program (FETP) trainees and graduates is highlighted; however, expanding on how their unique perspectives specifically contribute to understanding barriers and enablers could further enrich the narrative."

We further expand on the unique perspectives provided by FETP personnel in lines 84-87, in addition to the information previously provided in the Introduction section. We further clarified the rationale for reaching out to FETP personnel in lines 184-192.

"4. Methodological Rigor: More information on the validity and reliability of the survey instruments and interview protocols would be useful. Additionally, discussing any pilot testing of the instruments might help reassure readers of the methodological rigor."

We discuss and expand on pilot testing of the survey and interviews in lines 268-271 and lines 294-297, respectively. We note that although we can assess for content validity, as an exploratory, anonymous survey other measures of validity and reliability are not applicable. We cannot establish internal validity because unlike a psychometric assessment, in which questions are related and responses are expected to vary in the same direction, there is no empirical basis to expect the barriers and enablers examined in this study to vary together, making measures of internal validity like Cronbach’s alpha inapplicable here. As a unique survey that has no known antecedent in terms of both methodology and subject matter, there is no gold standard or otherwise that this survey can be assessed against to evaluate for external validity. Finally, as an anonymous survey, it is not conducive to having participants repeat the survey to assess for reliability. We hope that future surveys in the field will allow for a more standardized assessment of validity and reliability, especially as they are repeated across different environments.

"5. Ethical Considerations: The manuscript mentions ethical approvals and consent processes but could further discuss how potential ethical issues in dealing with sensitive outbreak data will be managed, especially in interviews."

This is an important point, and we thank you for pointing it out. We further elaborate on how we will redact any location or event information that could potentially be linked to interviewees or their countries of work in lines 400-403.

"6. Limitations: I appreciate the discussion of limitations; however, it might be beneficial to elaborate on how these limitations could affect the study's outcomes and the steps taken to mitigate them."

We further clarify the limitations' impact on study outcomes and how we intend to mitigate these limitations in lines 445-448 and lines 451-454, respectively.

"7. Implications for Policy and Practice: While the manuscript discusses the study's potential impacts, specific recommendations for policy-makers and public health officials based on anticipated findings could make the conclusions more actionable."

We elaborate on specific recommendations in lines 469-474.

Reviewer #3:

"Title – Appropriate

Abstract – Well presented. However, replace Discussion with Conclusion"

We replaced Discussion with Conclusion in the Abstract.

"Introduction – Statement of problem, magnitude of problem, rationale for the study and study objectives well presented.

Line 128 - Does this mean health care providers in public and private health facilities?"

We mean providers in both public and private health facilities. We’ve changed the language in line 134 to reflect this.

"Methods and analysis – Clearly presented. Specific study design not mentioned - Is it a descriptive cross-sectional study?"

We clarify in line 159 that this is a cross-sectional mixed methods study.

"Study setting, target population, sampling techniques and data management are well described

Line 155 - It is desirable to state where the piloting will be done."

We clarify in lines 269 and 295 where the piloting will be done.

"Line 250 - Please indicate the number of respondents targeted for the pilot survey. Also indicate geographical locations of the pilot survey."

We clarify in lines 269 and 295 the number of persons intended to pilot the instruments and where the piloting will be done.

"Line 323 - It is not clear if this will be Key Informant Interview or In-Depth Interview. It needs to be clearly stated."

We clarify in lines 149-153 and lines 163-167 that these will be in-depth interviews.

"Discussion – Inappropriate for proposal as a sub-section. Most of the content is Limitation of the study. This can be made the sub-section title."

We inserted a separate Limitations sub-heading for heightened reader clarification within the larger discussion around the impact of this study.

"References - Adequate"

Additional edits:

We provide more specific objectives in the Study question and objective section.

We would be happy to make any additional clarifications as needed.

Kind regards,

Amish Talwar, MD, MPH

---

## [Editor Report · Decision Letter 1]

13 Aug 2024

Investigating the barriers and enablers to outbreak reporting in the Asia-Pacific region: a mixed-methods study protocol

PONE-D-24-06613R1

Dear Dr. Talwar,

We’re pleased to inform you that your manuscript has been judged scientifically suitable for publication and will be formally accepted for publication once it meets all outstanding technical requirements.

Kind regards,

Olushayo Oluseun Olu

Academic Editor

PLOS ONE
---

## [Editor Report · Acceptance letter]

19 Aug 2024

PONE-D-24-06613R1 

PLOS ONE

Dear Dr. Talwar, 

I'm pleased to inform you that your manuscript has been deemed suitable for publication in PLOS ONE. Congratulations! Your manuscript is now being handed over to our production team.

Kind regards, 

on behalf of

Dr. Olushayo Oluseun Olu 

Academic Editor

PLOS ONE